# Swiss public attitudes to human cryopreservation

José Paulo Rodrigues dos Santos[1,2☉], Nadia X. Montazeri[1☉], Tijana Perović[ID][1], Emil F. Kendziorra[ID][1,3]*

**1** Tomorrow Biostasis, Berlin, Germany, **2** FH Aachen University of Applied Sciences, Aachen, Germany, **3** European Biostasis Foundation, Riehen, Canton of Basel-Stadt, Switzerland

☉ These authors contributed equally to this work.
* emil.kendziorra@ebf.foundation

## Abstract

Cryopreservation, or cryonics, is an experimental procedure that preserves individuals at cryogenic temperatures after legal death in the hope of future revival. Although Switzerland hosts the Schengen Area's first dedicated whole-body human cryopreservation facility, public sentiment toward the practice has remained largely unexamined. This exploratory survey of 249 Swiss adults assessed awareness, ethical views, and openness to cryopreservation. Results show broad support for individual autonomy, with most respondents endorsing the right to choose cryopreservation when performed to high medical standards (86.7%) and not supporting legal restrictions (83.5%). While personal interest was in the minority, nearly one in five respondents (20.1%) reported active interest or intent to sign up. Openness to cryopreservation appears driven more by values such as life-extension preference and prior exposure than by demographics. These findings provide the first empirical snapshot of Swiss public opinion on cryopreservation, highlighting a largely permissive public stance and suggesting considerable engagement with the topic.

## Introduction

Cryopreservation, also known as cryonics or biostasis, is an experimental procedure in which a legally deceased person is cooled to −196 °C, typically in liquid nitrogen, while avoiding ice crystal formation, in the hope that future medical advances may enable revival and the alleviation of old age and curing of disease. The process begins as soon as the heart stops, but at the earliest after legal pronouncement, and uses medical-grade antifreeze to protect tissues during cooling [1].

Despite decades of development, philosophical debates [2–4] and increasing media coverage, empirical research on public sentiment toward cryopreservation remains limited. The field continues to attract curiosity [5], while ethical, philosophical, and scientific questions remain widely debated. Fewer than 1,000 people worldwide have been cryopreserved [6], though interest appears to be growing: about 3,100 individuals in the U.S. have made arrangements for future preservation [1]. The same

**Data availability statement:** All relevant data are within the paper and its Supporting Information files.

**Funding:** The author(s) received no specific funding for this work.

**Competing interests:** TP is an employee of Tomorrow Biostasis GmbH. EK is a shareholder and CEO of Tomorrow Biostasis GmbH, a biostasis provider, President of the Board of the European Biostasis Foundation, a non-profit research foundation, a shareholder and director at Oxford Cryotechnology, Inc., a cryopreservation research organization, and a board member at CryoDAO, a Swiss research association. There are no patents or products in development associated with this research to declare. Tomorrow Biostasis provides its clients with already marketed products (human cryopreservation), which are tangentially related to the topic of the paper, but the paper offers no direct benefit to the product. Tomorrow Biostasis markets human cryopreservation services and holds intellectual property related to cryopreservation technology. The company had no role in the study's design, execution, analysis, or publication. The survey was anonymous and conducted independently of the company's customer base, and no company marketing materials were used in recruitment. The authors EK and TP confirm that their affiliation does not alter their adherence to PLOS ONE policies on sharing data and materials.

U.S. survey found that 75% of respondents had heard of cryopreservation, 20% expressed interest, and only 6% had decided to be cryopreserved. In Europe, a 2014 German survey (n = 1,000; age 16–69) reported that 47% were aware of cryopreservation and 22% were open to the idea, though some respondents expressed ethical or philosophical opposition [7].

No equivalent data exists for Switzerland, even as the country gains prominence in the cryopreservation landscape. In 2022, the first dedicated whole-body human cryopreservation facility in the Schengen Area, for the long-term storage of cryopreserved people opened in Switzerland. As of 2023, the first individuals are in storage at the facility [8]. Switzerland's political stability makes it an appealing location for long-term cryogenic storage [9].

To address this gap, we conducted the first survey of Swiss public attitudes toward cryopreservation. We asked a representative sample about their familiarity with, interest in, and ethical views on cryopreservation, offering new insight into how the practice is perceived in a key European context.

## Methodology

### Participants and recruitment

We recruited participants through Positly [10], a research recruitment service based in New York, USA, between March 11 and April 17, 2025. Positly also managed the consent process. All participants were at least 18 years old, took part voluntarily, and remained anonymous. Each received monetary compensation for their time. The data were accessed for research purposes between April 18 and August 22, 2025. At no point did the authors have access to information that could identify individual participants, during or after data collection.

We collected 283 responses through an online survey built with Guided Track [11]. Thirty-two responses completed in under three minutes were excluded, as internal testing indicated that this was the minimum time required to provide valid responses. This left 251 responses for analysis. One additional respondent was excluded for not residing in Switzerland and another for being under 18, resulting in 249 valid responses.

All participants were Swiss residents, and we used Swiss census data to ensure the sample was representative by age and gender. The survey was conducted in German and English to partially account for Switzerland's multilingual population. In total, 52 responses were in English and 231 in German. Of the 34 excluded responses, 6 were in English and 28 in German.

### Ethics declaration

This study is exempt from the full scope of Switzerland's Human Research Act (HRA, Art. 2), as it does not investigate human diseases, bodily functions, or public health outcomes, nor does it involve any form of physical or psychological intervention. The research consists exclusively of purely observational, anonymous surveys, which fall outside the HRA's regulatory remit.

Data collection was conducted in full compliance with the Swiss Federal Act on Data Protection (FADP). The study design ensured that no personally identifiable information was collected or retained at any point; survey responses could not be attributed to individual participants under any circumstances. Data was gathered via Positly, a GDPR-compliant platform whose data protection standards align with the requirements of the Swiss FADP, providing an additional layer of regulatory assurance.

Informed consent, data protection information, and the terms and conditions of participation were provided to all subjects through the Positly online platform prior to enrollment (https://www.positly.com/terms/). By registering, participants confirmed they were at least 18 years of age and agreed that data collected during the study may be used for scientific research purposes. Participants were explicitly informed that participation is entirely voluntary and that financial compensation is provided upon completion of the survey. They retained full discretion over whether to accept an invitation and engage with the study at any stage.

## Survey questions

The survey began with the following explanatory text, also shown again after the demographic questions:

> "Please share your perspective on an experimental practice of emergency medicine known as Biostasis/ Cryopreservation or Cryonics.
>
> Please read carefully: Biostasis/Cryopreservation is a method of 'freezing' people who have just died with the aim of potentially 'reviving' these persons in the future - with the help of more advanced medicine.
>
> In an ideal case, a cryopreservation team begins the procedure at the moment the patient's heart stops.
>
> They quickly restart circulation as they lower the body's temperature to just above freezing, apply medical-grade anti-freeze, then bring the body's temperature to -196 degrees Celsius for long-term storage in liquid nitrogen.
>
> The oldest cryopreservation provider is a non-profit that has been operating continuously for 53 years. Hypothetically, a well-preserved brain may have enough information for future scientists to restore memories and consciousness."

The full list of survey questions and answers can be found in S1 Table.

## Data processing and analysis

We exported the data from GuidedTrack as CSV and used Python libraries (pandas [12], NumPy [13], scipy.stats [14], seaborn [15], statsmodels [16], and matplotlib [17]) for preparation and analysis. Responses on a Likert-type scale were converted to numeric values ranging from –2 to +2 for statistical analysis. For example, a five-point scale of "Strongly Agree," "Agree," "Neither Agree nor Disagree," "Disagree," and "Strongly Disagree" was coded as +2, +1, 0, –1, and –2, respectively.

We performed descriptive (distribution of responses) and inferential analyses. In particular, we assessed associations between demographic factors and respondents' opinions toward cryopreservation. To control for multiple Spearman correlation comparisons, we applied the Bonferroni method to all p-values, multiplying each by the number of total tests.

We also conducted a chi-square test of independence to examine the association between openness to cryopreservation and information-seeking behavior. Effect size was measured using Cramer's V.

## Results

Our primary objective was to assess how Swiss respondents perceive cryopreservation—whether they see it as a personal decision, something to be discouraged, or otherwise. We examined public sentiment toward cryopreservation for themselves and family members, their legal understanding of the practice, and their overall openness to the idea.

## Demographics of respondents

Of the 249 responses, 52.2% were female and 47.8% were male. The median age was 46 years, with a range of 18–77. These figures align with Swiss census data for age and gender. The median income was 73,213.36 CHF, slightly below the national median [18]. Of respondents, 44.2% were married, 42.6% had never married, and 13.3% were divorced or widowed. Just over half (54.6%) had children, and 35.8% held a bachelor's degree or higher.

## Sentiments toward cryopreservation

Our findings indicate that 79.5% of respondents believe cryopreservation, when performed professionally, should be considered an individual's decision rather than a violation of social or ethical norms. If cryopreservation meets high standards of medical professionalism, 86.7% agree it should be regarded as a matter of personal choice (Fig 1).

Survey results show that there was no widespread interest in making cryopreservation illegal, therefore changing its current status (Fig 2). A total of 38.9% respondents explicitly disagreed or strongly disagreed and 44.6% expressed neutrality. Only a minority of 16.4% agreed or strongly agreed with making it illegal.

While most respondents do not believe that solving the problem of death through cryopreservation is worth pursuing, a relevant minority, 30.5%, said they would like to see what the future holds, and 25.7% stated they would choose to live indefinitely if possible (Fig 3). Additionally, 19.7% reported being frightened by death, 25.7% found cryopreservation exciting, and 15.7% believed there is a good chance it will eventually work. Moreover, 23.7% expressed interest in living another 100 youthful years if they could (Fig 3).

Despite these attitudes toward death, end-of-life decisions are still shaped by more immediate concerns. Cost remains the main factor in choosing death care arrangements for most respondents (48.6%), followed by family wishes (45%). This finding aligns with previous reports (1998) [19,20]. Nonetheless, 20.1% of respondents had either signed up for cryopreservation or expressed interest in doing so.

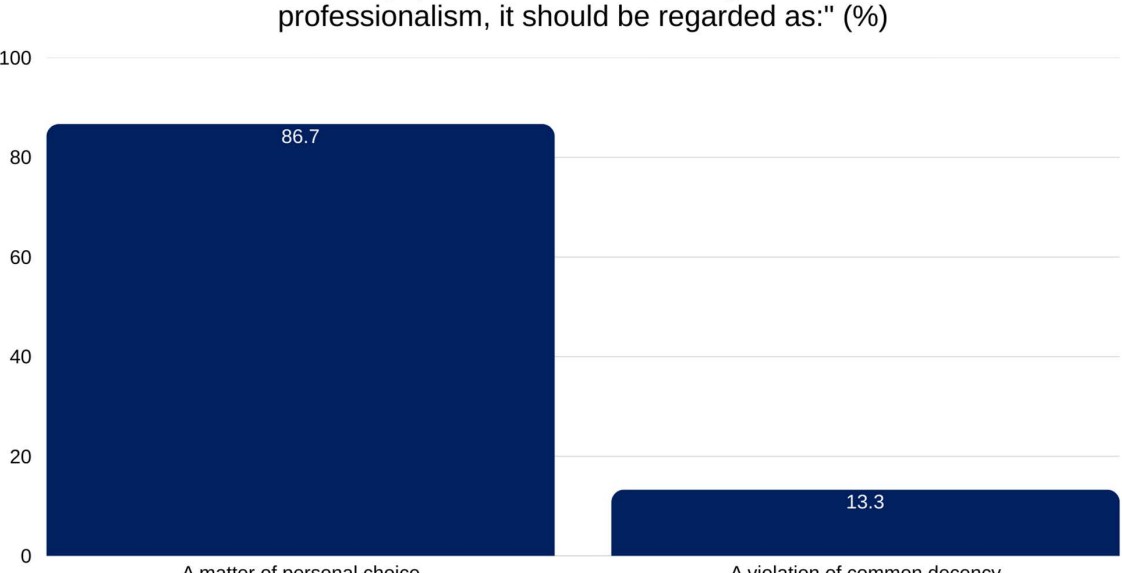

**Fig 1. Distribution of responses to the statement "If cryopreservation meets a high standard of medical professionalism, it should be regarded as:".** A large majority (86.7%) viewed it as a matter of personal choice, while 13.3% considered it a violation of common decency. Percentages reflect total valid responses.

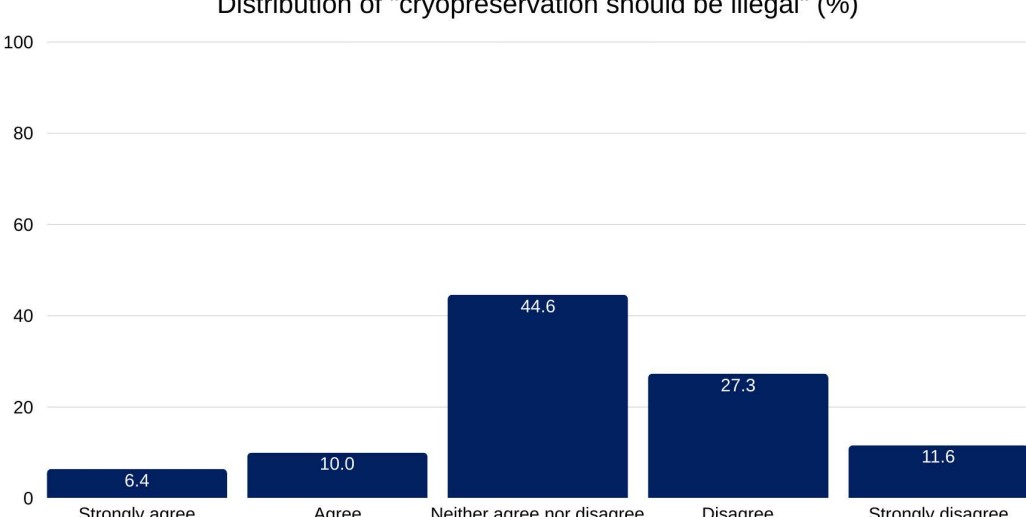

**Fig 2. Distribution of responses to the statement "Cryopreservation should be illegal. "**. A plurality of respondents (44.6%) selected "Neither agree nor disagree," while 27.3% disagreed and 11.6% strongly disagreed. A smaller proportion agreed (10.0%) or strongly agreed (6.4%) with the statement. Percentages are shown as a proportion of total valid responses.

A combined 75.0% of respondents would either support or remain neutral toward a family member's decision to pursue cryopreservation, reinforcing the notion of broad acceptance—even if not personal endorsement (Fig 4).

## Attitudes and associations

Beyond overall attitudes, the data reveal notable relationships between demographic factors, personal beliefs, and openness to cryopreservation. Fig 5 displays the Spearman correlation matrix between key variables with Bonferroni-corrected p-values. A moderately strong positive correlation emerged between openness to cryopreservation and the belief that cryopreservation has a good chance of working ($\rho_s = 0.42$, $p < 0.001$). Openness to cryopreservation was also weakly but significantly correlated with Death Frightens Me ($\rho_s = 0.21$, $p = 0.039$). No noteworthy correlations were found between openness to cryopreservation and age, income, or the belief in wanting to live much longer or even forever.

Age was a weak predictor for several other key variables. A negative correlation appeared between age and "Death Frightens Me" ($\rho_s = -0.21$, $p = 0.033$), consistent with Rasmussen and Brems (1996) [21]. Similarly, the appeal of living indefinitely in good health declined with age, showing a significant negative correlation between Age and agreement with the statement, *"If I could live indefinitely in good health, even 'forever,' I would"* ($\rho_s = -0.23$, $p = 0.011$). Age also showed a significant negative correlation with the belief that cryopreservation has a good chance of working ($\rho_s = -0.22$, $p = 0.019$). These results align with the Less Wrong survey [22], which found that older people are less likely to be interested in cryopreservation.

Finally, while both fear of death and the wish to live forever showed a positive—but non-significant—correlation with openness to cryopreservation, they were not significantly correlated with each other or with the belief that cryopreservation has a good chance of working. Likewise, Income did not correlate significantly with any other variables in the matrix (Fig 5).

The acceptance and potential viability of cryopreservation are closely linked to how people engage with and perceive the field. For example, there is a positive correlation ($\rho_s = 0.32$, $p = 0.001$) between finding cryopreservation "exciting" and expressing a desire to live another 100 years or more.

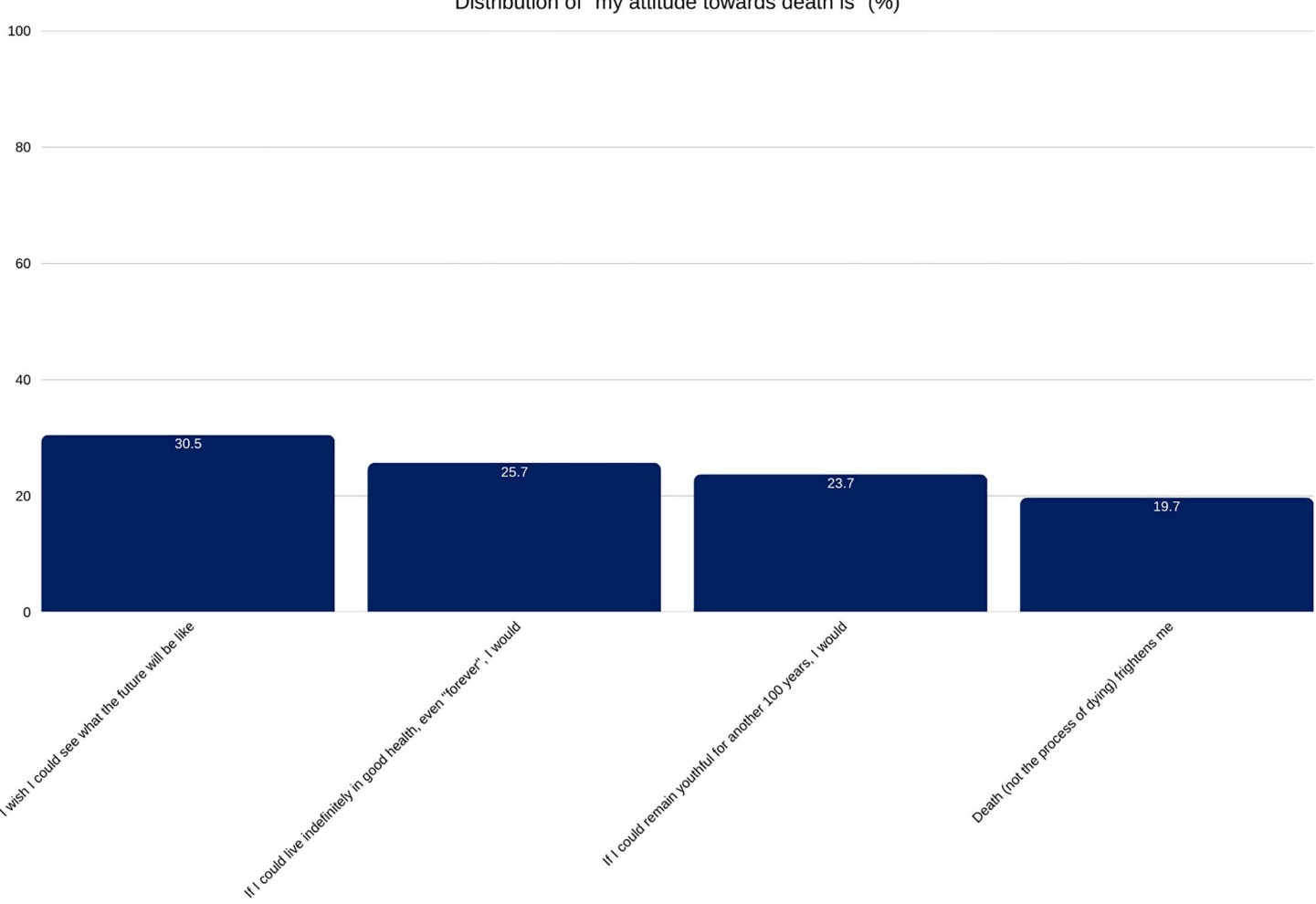

Distribution of "my attitude towards death is" (%)

- I wish I could see what the future will be like — 30.5
- If I could live indefinitely in good health, even "forever", I would — 25.7
- If I could remain youthful for another 100 years, I would — 23.7
- Death (not the process of dying) frightens me — 19.7

**Fig 3. A vertical bar chart titled "Attitude Towards Longevity."** " It visually compares the percentage of total survey participants who agreed with two different statements about extending life. Options were part of a multi-select questionnaire.

Active engagement with the topic is also associated with a more positive view of its efficacy. A moderate correlation exists between having sought information about cryopreservation and believing it has a good chance of working ($\rho_s = 0.319$, $p = 0.001$). Seeking information also correlates with finding some companies in the field trustworthy ($\rho_s = 0.402$, $p < 0.001$).

There is a strong association between active information-seeking and openness to cryopreservation ($\chi^2 = 54.948$, $df = 2$, $p < 0.001$, Cramer's V $= 0.470$).

Only 10% of participants correctly identified the reason vitrification [23], the primary technology used in most cryopreservation cases, is employed, and only 3.6% correctly identified the typical cost (USD 180,000–220,000).

The analysis also examined how seeking information and familiarity with cryopreservation relate to attitudes toward the practice. Fig 6 shows Spearman's ρ values with Bonferroni-corrected p-values. The strongest correlations appear with the "Sought Info on Cryopreservation" variable: individuals who sought information were more likely to be personally open to cryopreservation, support it for family members, consider it exciting, and believe it has a good chance of success. All of

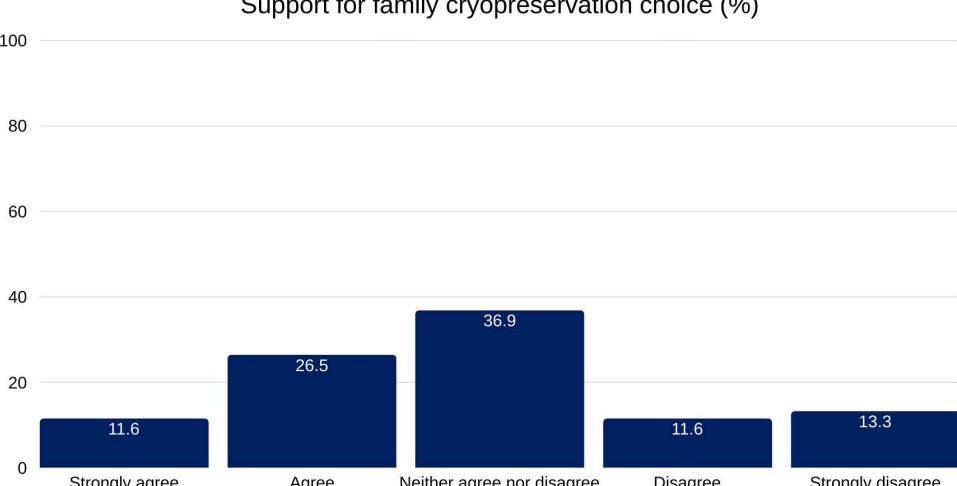

**Fig 4. Distribution of responses to the statement "I would support a family member's decision to be cryopreserved.** " The most common response was neutral (36.9%), followed by agreement (26.5%). A smaller proportion expressed strong disagreement (13.3%), disagreement (11.6%), or strong agreement (11.6%). Percentages reflect total valid responses.

these relationships were statistically significant (p < 0.05), reinforcing earlier findings that active engagement correlates with positive perceptions and increased trust in the field.

Age showed a significant negative correlation with both cost estimates—how much respondents guessed cryopreservation would cost—and information-seeking, meaning how familiar they were with the practice. Older respondents tended to give lower cost estimates and were less likely to have sought information. In contrast, the number of cryopreserved individuals known and understanding of vitrification showed weaker and mostly non-significant correlations, suggesting that direct exposure and technical knowledge remain limited in the general population (Fig 6).

## Discussion

This study offers an overview of Swiss perceptions of cryopreservation, revealing a nuanced public sentiment that strongly leans toward acceptance and personal autonomy. The findings indicate considerable social acceptance, with openness and neutrality prevailing over aversion. However, public knowledge remains limited, and personal interest is confined to a notable minority.

### Autonomy and openness

A key finding is the overwhelming public sentiment in favor of keeping cryopreservation legal, with 83.6% of respondents expressing no support for legal restrictions nor making a change from its current status. Support is reinforced by 86.7% of respondents who believe cryopreservation, when performed to high medical standards, should be a matter of personal choice. Together, these findings highlight a strong cultural emphasis on autonomy, consistent with broader European values that uphold individual rights in matters of life and death, provided they do not harm others [24].

The moderately strong correlation between seeking information about cryopreservation and finding some companies in the field trustworthy ($\rho_s = 0.402$, p < 0.001) suggests that those who investigate the topic are more likely to view certain organizations in the industry as reputable.

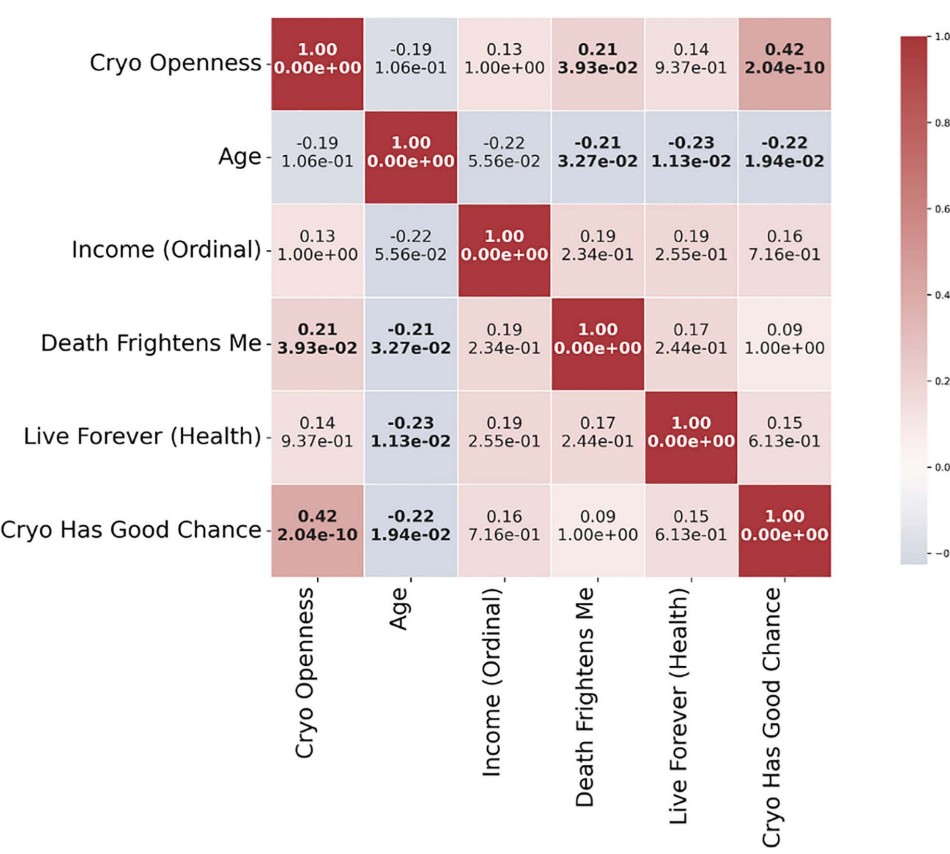

**Fig 5. Spearman correlation matrix showing coefficients (top) and Bonferroni-corrected p-values (bottom) for all variable pairs.** Red indicates positive correlations, while blue indicates negative correlations. The color bar on the right shows the scale of the correlation coefficients.

Similarly, the positive correlation between seeking information and openness to cryopreservation ($\chi^2 = 54.948$, df $= 2$, p $< 0.001$, Cramer's V $= 0.470$) raises the question of directionality. It is unclear whether initial openness prompts individuals to seek more information, or whether seeking information fosters greater openness. A topic that should be further elucidated in follow-on surveys.

## Comparison with existing research: Awareness and interest in cryopreservation

The Swiss survey results on awareness and interest in cryopreservation align broadly with findings from the U.S. and Germany, offering insight into public perception across Western countries.

## Awareness of cryopreservation

In Switzerland, 52.7% of respondents had heard of cryopreservation—placing the country between the U.S. (75%) and Germany (47%) in awareness levels. Higher awareness in the U.S. may stem from its longer history with the practice and the presence of long-established organizations. Switzerland's slightly higher awareness than Germany does not imply causation, but it may reflect more recent developments, greater media exposure, and the 11 years since the 2014 study by Kaiser [7].

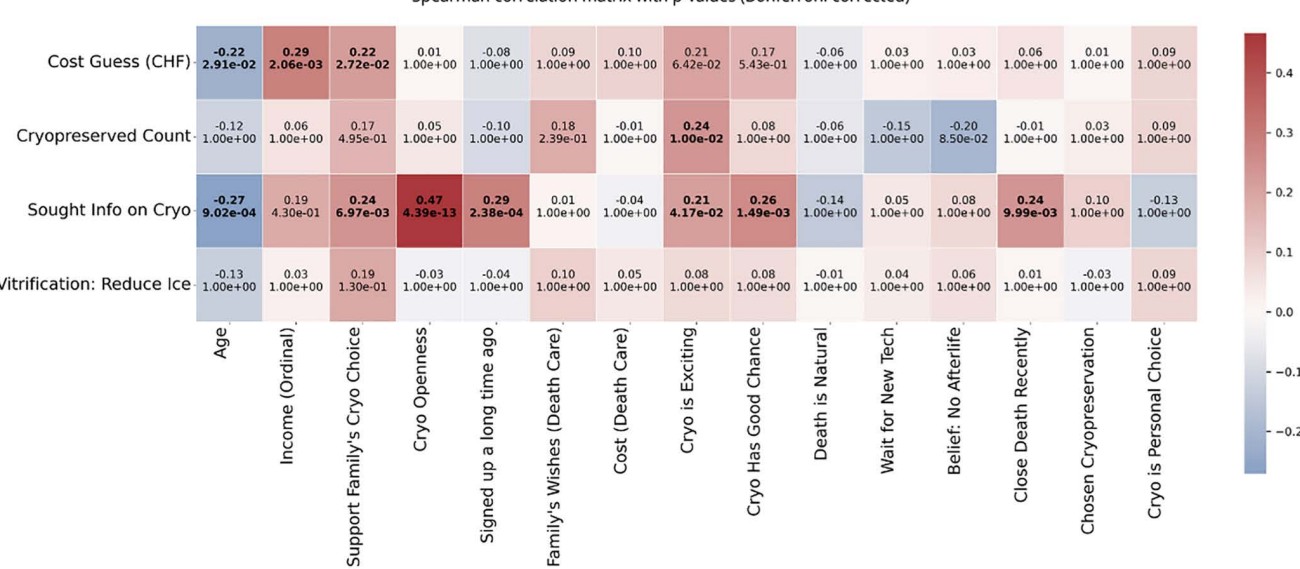

**Fig 6. Spearman correlation matrix between key survey variables and four indicators of cryogenics awareness (Y-axis).** The matrix includes perceived cost (in CHF), number of cryopreserved individuals known, self-reported information-seeking behavior, and understanding of vitrification. Correlation coefficients are shown alongside Bonferroni-corrected p-values. Red indicates positive correlation, blue indicates negative correlation. Statistically significant correlations ($p < 0.05$) after correction are bolded.

## Interest in cryopreservation

Despite differences in awareness, interest in cryopreservation is remarkably consistent across the U.S., Germany, and Switzerland. In the Swiss survey, 20.1% of respondents expressed interest or were already signed up—almost identical to the 20% in the U.S. and 22% in Germany. This suggests a stable baseline of interest regardless of how well-known the concept is. Even in populations less familiar with cryopreservation, a similar proportion finds it appealing once exposed. However, the proportion considering cryopreservation remains far greater than the proportion actually enrolling, indicating potential barriers to taking this step.

Our findings show a link between general openness to cryopreservation and confidence in its future success ($\rho_s = 0.42$, $p < 0.001$). This suggests a self-reinforcing dynamic in which interest and optimism are closely intertwined: those more receptive to the concept are also far more likely to believe it will eventually work. The trend is especially pronounced among younger respondents, with age negatively correlated with the belief that cryopreservation has a good chance of succeeding ($\rho_s = -0.22$, $p = 0.019$). Younger individuals appear more optimistic about its feasibility.

## Openness–awareness dynamic

A moderately strong association exists between active information-seeking and openness to cryopreservation ($\chi^2 = 54.948$, df = 2, $p < 0.001$, Cramer's V = 0.470), one of the strongest correlations in our data. The Openness–Awareness Dynamic considers whether awareness leads to openness or openness leads to awareness. One interpretation is that openness comes first: individuals intrigued by cryopreservation may seek more information. Supporters in this group may seek information that confirms their stance, focusing on positive accounts and scientific arguments.

An alternative interpretation is that awareness precedes openness: initially neutral or skeptical individuals could become more receptive as they learn about the science, philosophy, and procedures behind cryopreservation. In practice,

the relationship may operate as a cycle—some openness prompts information-seeking, which increases openness, which in turn motivates deeper investigation—creating a reinforcing loop that makes a single point of origin hard to define.

This interpretation is consistent with Bandura's reciprocal determinism [25], which holds that behavior, personal factors (such as openness), and environmental exposure (such as awareness) influence one another in a continuous cycle rather than in a single direction.

Research in framing effects suggests that how a novel technology is presented (medical versus technical curiosity) shapes cognitive and affective responses [26,27]. Surveys can be conducted anonymously, online, or in-person and this context is also a potential source of bias (e.g., social desirability bias) [28].

## Coping vs pro-longevity arguments

Our study found a weak but significant correlation between fear of death and openness to cryopreservation ($\rho_s = 0.21$, $p = 0.039$), in contrast to a major U.S. survey [1] where fear of death was not a significant motivator. This suggests that some individuals may view cryopreservation as a solution-oriented way of coping with death anxiety.

At the same time, our findings indicate that interest in cryopreservation may be driven not only by fear of death but also by a desire to extend life, with both showing weak to moderate correlations. Fear of death was slightly more strongly associated with openness to cryopreservation than the desire for life extension, yet the latter remains a relevant factor. In a multiple-select question, 25.7% of respondents chose to live indefinitely, if possible, while 23.7% selected an additional 100 youthful years. For these individuals, the appeal seems rooted in the opportunity to experience more life, and their openness to cryopreservation may reflect a positive, life-affirming motivation rather than simple avoidance of death. Notably, a few more respondents selected living indefinitely than living an additional 100 youthful years, suggesting a degree of inconsistency in responses.

## Openness to cryopreservation as a philosophical stance

The absence of impactful demographic effects on openness to cryopreservation, combined with the measured influence of fear of death ($\rho_s = 0.21$, $p = 0.039$) and finding cryopreservation exciting ($\rho_s = 0.32$, $p < 0.001$), suggests that motivation for cryopreservation in our sample is not strongly linked to socioeconomic status (e.g., income) or life stage (e.g., age).

This pattern points to openness to cryopreservation as an expression of broader psychological or philosophical orientations toward life preservation, rather than factors tied to a specific demographic profile.

Although cryopreservation involves significant costs, no substantial correlation between income and openness emerged. This may indicate that current financial capacity is not the main driver of openness, though further research is needed to clarify the underlying reasons. Likewise, the stability of openness to cryopreservation across age groups suggests that interest in cryopreservation is not primarily a late-life reaction, but is present across the lifespan. Although not assessed here, it has been suggested that openness correlates with familial/communal acceptance of the procedure and accompanying lack of traditional funeral and mourning rituals [20]. However, matters of moral intuition – e.g., purity concerns or mortality salience – may interact with framing to shape attitudes towards cryonics, consistent with German & Tretter [29].

## Ambivalence and neutrality

A notable finding is the link between a minority having personal interest in cryopreservation and high social permissiveness toward it. While much of the Swiss public is disengaged—often reporting no awareness (47.3%) or no intent to sign-up (79.9%)—most respondents (86.7%) believe that, if performed professionally, cryopreservation should remain a matter of personal choice. Only a small minority, 16.4%, support making it illegal.

Many respondents adopt a neutral or ambivalent stance, neither fully endorsing nor rejecting the practice. This pattern also appeared in Germany in Kaiser's 2014 study [7]. People may hesitate to take a definitive position on a practice they understand poorly, especially given its technical and ethical complexity. Only 10% of participants correctly identified the reason vitrification [23], the primary technology used in cryopreservation, is employed in most cases, and just 3.6% knew the exact financial cost of cryopreservation.

This neutrality may also reflect a deeper respect for individual autonomy, a value embedded in Swiss culture [30,31]. The prevailing attitude might be summed up as: *"I would not choose this for myself, but it is not my place to prevent others from choosing it."* In this way, the Swiss public seems to reserve judgment, creating a permissive space in which cryopreservation can exist without full endorsement.

Finally, the fact that 20.1% of respondents have either signed up for cryopreservation or expressed interest in doing so is significant. While many cite cost (48.6%) and family wishes (45%) as primary factors in end-of-life decisions, the level of interest in cryopreservation suggests a high openness to innovative end-of-life choices in Swiss society.

### Limitations and future directions

This study relies on self-reported data, which may be influenced by social desirability bias. While the demographic profile aligns with Swiss census figures, larger and more diverse samples could offer a more comprehensive view of public sentiment. Future research should also explore why many respondents adopted a neutral stance.

A longitudinal design could track how attitudes evolve over time—particularly among younger respondents, who showed greater optimism—and clarify the cyclical relationship between information-seeking and openness.

Although our sample size (n = 249) is modest, it is adequate for an exploratory study and provides a valuable first look at Swiss attitudes toward cryopreservation. In addition, a further study could be conducted in Switzerland, including the country's other native languages: French, Italian, and Romansh.

### Conclusion

In Switzerland, cryopreservation is widely regarded as a matter of personal choice, with strong support for individual autonomy and little appetite for legal restrictions. While most respondents remain neutral, the lack of opposition and reported family support suggest cautious acceptance.

Openness to cryopreservation appears driven more by individual values than by demographics. Fear of death, optimism, and access to information shape attitudes and may reinforce one another over time.

With 20.1% expressing interest or intent to sign up, cryopreservation is emerging as a considered option rather than a distant idea. The field's future may depend on the large, cautiously neutral segment of the population, and on ensuring clear professional standards that align with Switzerland's commitment to informed, individual choice.

### Supporting information

**S1 Table. Complete data-frame.** This table provides the complete anonymized dataset underlying the analyses, including survey language, time spent on the survey (minutes), coded position, project, country, age (years), marital status, children, education level, and responses to survey items on public opinion regarding cryonics.
(XLSX)

### Author contributions

**Conceptualization:** Nadia X. Montazeri, Emil F. Kendziorra.

**Data curation:** Nadia X. Montazeri.

**Formal analysis:** José Paulo Rodrigues dos Santos, Nadia X. Montazeri.

**Funding acquisition:** Emil F. Kendziorra.

**Investigation:** Nadia X. Montazeri.

**Methodology:** José Paulo Rodrigues dos Santos, Nadia X. Montazeri, Emil F. Kendziorra.

**Project administration:** José Paulo Rodrigues dos Santos, Nadia X. Montazeri, Emil F. Kendziorra.

**Resources:** Emil F. Kendziorra.

**Software:** José Paulo Rodrigues dos Santos, Nadia X. Montazeri.

**Supervision:** Emil F. Kendziorra.

**Validation:** Emil F. Kendziorra.

**Visualization:** José Paulo Rodrigues dos Santos.

**Writing – original draft:** José Paulo Rodrigues dos Santos, Nadia X. Montazeri.

**Writing – review & editing:** Emil F. Kendziorra, Tijana Perović.

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
