## [Decision Letter · Decision Letter 0]

8 Sep 2025

PONE-D-25-45890Cryopreservation sentiment in SwitzerlandPLOS ONE

Dear Dr. Kendziorra,

Thank you for submitting your manuscript to PLOS ONE. After careful consideration, we feel that it has merit but does not fully meet PLOS ONE’s publication criteria as it currently stands. Therefore, we invite you to submit a revised version of the manuscript that addresses the points raised during the review process.

Please see the reviewers comments below, covering the required changes.

We look forward to receiving your revised manuscript.

Kind regards,

Barry L. Bentley, Ph.D.

Academic Editor

PLOS ONE

Journal Requirements:

4. We note that there is identifying data in the Supporting Information file <S1 Complete Dataframe.xlsx>. Due to the inclusion of these potentially identifying data, we have removed this file from your file inventory. Prior to sharing human research participant data, authors should consult with an ethics committee to ensure data are shared in accordance with participant consent and all applicable local laws.

-Location data

Reviewers' comments:

Reviewer's Responses to Questions

**Comments to the Author**

1. Is the manuscript technically sound, and do the data support the conclusions?

Reviewer #1: Yes

Reviewer #2: Yes

2. Has the statistical analysis been performed appropriately and rigorously? 

Reviewer #1: Yes

Reviewer #2: Yes

3. Have the authors made all data underlying the findings in their manuscript fully available?

Reviewer #1: Yes

Reviewer #2: Yes

4. Is the manuscript presented in an intelligible fashion and written in standard English?

Reviewer #1: Yes

Reviewer #2: Yes

5. Review Comments to the Author

Reviewer #1: This is a well-done survey on an important topic. It is useful data and overall written in a reasonable way. It is interesting to see the correlations and overall it is consistent with previous data.

My one main concern is regarding the conclusion here: "Survey results show that most participants were against making cryopreservation illegal (Fig 2). A total of 83.5% of respondents either opposed or remained neutral toward a ban, with 38.9% explicitly disagreeing or strongly disagreeing and 44.6% expressing neutrality. Only 16.4% agreed

or strongly agreed with making it illegal."

I don't think it is accurate to say on the basis of these results that 83.5% of people are *against* making it illegal, if they have responded that they are neutral to that question. I think it is more accurate to say that 38.9% disagree with this, while 44.6% are neutral. It also would be accurate to say that only a minority (16.4%) think that it should be made illegal. So I think it would be fair to conclude that there is no expressed widespread interest in making it illegal, or making a change from its current status. This is an important conclusion of the survey and I think that the language describing this result should be precise throughout the manuscript in order to be accurate.

Also, some of the citations could be formatted better. For example, some of them seem to have authors, but the authors are not listed in the citation. It might also be useful to have a few citations in the beginning to the general topic of cryopreservation/biostasis/cryonics, so that readers know specifically what you are referring to.

One minor point regarding: "This suggests that some individuals may view cryopreservation as a way of coping with death anxiety". I would specifically describe this as a "problem-solving way of coping", rather than just a way of coping. Because theoretically, it's not just about feeling better, which is sometimes the connotation of the word "coping", but rather attempting to solve the problem of death.

Reviewer #2: dos Santos, Montazeri and Kendziorra report an exploratory online-survey of swiss residents on attitudes towards cryonics. By showing support for legality of cryonics in Switzerland and replicating results from the German population by Kaiser at al. 2014, the present study is of general interest for capturing temporal, geographical and cultural determinants of medical and end-of-life preferences. It therefore constitutes a valuable contribution to the cryonics literature. I support publication provided the issues below are addressed:

1. Line 1: The title “Cryopreservation sentiment in Switzerland” is too broad. “Cryopreservation” covers embryos, gametes, organs, etc. The survey is specifically about human cryopreservation/cryonics. “Stated attitudes” would be more precise than “Sentiment”. My suggestion for an alternative title is “Swiss public attitudes to cryonics”.

2. Line 8 and 34: The authors describe the 2022 EBF cryonics facility as the first in Europe. The authors should modify this claim, due to the KrioRus facility established in 2005 in geographical Europe. For instance, the authors might replace “Europe” with “Schengen Area”.

3. Line 26: The authors should discuss and include references to at least some landmark publications on cryonics in bioethical journals, including Shaw Bioethics 2009, Moen JME 2015, Thau Bioethics 2020, as well as more recent article like Sauchelli Sci Eng Ethics 2024, Shao Int J Phil Stud 2025, German and Tretter Neuroethics 2025

4. Figure 1: The authors might consider representing this data as a pie chart or a diverging stacked horizontal bar chart.

5. Line 128: The authors may also relate their findings regarding family wishes to Hillenbrink R, Wareham CS. Mourning the frozen: considering the relational implications of cryonics. J Med Ethics. 2024

6. Line 254 and Line 277: If successful, cryonics could have pronounced benefits. It is therefore of general interest to elucidate the underlying psychological and sociological causes for its appeal to the general population in different framings. The manuscript provides some engagement with this topic by pointing to Bandura's reciprocal determinism, but the authors should extend their discussion of potential causal effects. This includes the framing as a medical treatment in the utilized vignette, potential framing effects of an anonymous online survey vs., e.g., person-to-person and public conversations. The authors should relate their results to Laakasuo et al. 2018 https://doi.org/10.1057/s41599-018-0124-6 and 2021 https://doi.org/10.1016/j.paid.2021.110731 and German and Tretter 2025 https://doi.org/10.1007/s12152-025-09584-7

7. Please clarify competing interests, including affiliations and equity stakes related to Tomorrow Biostasis, the European Biostasis Foundation, and other pertinent entities discussed in the manuscript.

6. PLOS authors have the option to publish the peer review history of their article (what does this mean?). If published, this will include your full peer review and any attached files.

Reviewer #1: No

Reviewer #2: **Yes:** Alexander German

---

## [Author Response · Author response to Decision Letter 1]

24 Feb 2026

Dear Dr. Barry L. Bentley,

We thank you and the reviewers for the constructive comments. We have revised the manuscript thoroughly and provide a detailed, point-by-point response in the attached document "Response to Reviewers".

Major revisions include:

Title change

Adapted the description of Figure 2 results section

Expanded key discussion points brought up by Reviewer 2

Expanded list of references added to introduction and discussion

Added competing interests

We believe the manuscript is substantially improved and hope it is now suitable for publication in PLOS ONE.

Sincerely,

Emil Kendziorra

---

## [Editor Report · Decision Letter 1]

11 Mar 2026

PONE-D-25-45890R1Swiss public attitudes to human cryopreservationPLOS One

Dear Dr. Kendziorra,

Thank you for submitting your manuscript to PLOS ONE. After careful consideration, we feel that it has merit but does not fully meet PLOS ONE’s publication criteria as it currently stands. Therefore, we invite you to submit a revised version of the manuscript that addresses the points raised during the review process. Specifically, could you please address the following:

(1) On the previous request concerning identifying data, please remove the fields in the supporting data containing exact ages and retain only the age bins.

(2) Provide further detail in the main text on the consent process to demonstrate that informed consent was appropriately obtained from participants.

(3) Consistent with best practice and the WMA’s Declaration of Helsinki, I would reiterate the request for the study to be reviewed by an independent ethics panel, or provide clarification if such independent oversight has already taken place, in order to confirm that relevant ethical and legal requirements were followed (e.g. GDPR). Please feel free to contact me if you require advice on what is required for this.

(4) Ln 18 & 45-46: first dedicated cryopreservation facility: Thank you for clarifying the geographical scope. Along with Russia, Alcor operated a UK facility in the 1980s, so the change to clarify Schengen is appropriate. However, in line with Reviewer 2’s comment, the wording remains imprecise. The term “cryopreservation facility” is very broad and encompass a wide range of facilities (e.g. tissue banks, seed storage, brain banks, reproductive facilities, etc.). Please clarify in the text that the facility being discussed is specifically intended for whole-body human cryopreservation, as this is the important distinction.

(5) Unaddressed reviewer comment: Reviewer 2 suggested that the authors relate their findings regarding family wishes to the discussion in Hillenbrink & Wareham. This comment was not addressed in the response to reviewers. Please respond to this point and, if appropriate, incorporate relevant discussion into the manuscript.

We look forward to receiving your revised manuscript.

Kind regards,

Barry L. Bentley, Ph.D.

Academic Editor

PLOS One
---

## [Author Response · Author response to Decision Letter 2]

18 Mar 2026

Dear Editor,

Thank you for the opportunity to revise our manuscript. We have addressed all comments from the reviewers and have adjusted the manuscript accordingly. Please find our point-by-point responses below.

Comments from the Academic Editor:

Response: We have adjusted the formatting, including figure naming and author affiliations, to meet the guidelines.

2. PLOS requires an ORCID iD for the corresponding author in Editorial Manager on papers submitted after December 6th, 2016.

Response: We have added the corresponding author’s ORCID ID in Editorial Manager.

3. Please amend your list of authors on the manuscript to ensure that each author is linked to an affiliation. Authors’ affiliations should reflect the institution where the work was done.

Response: We have amended the authors list and affiliations.

4. We note that there is identifying data in the Supporting Information file <S1 Complete Dataframe.xlsx>. Due to the inclusion of these potentially identifying data, we have removed this file from your file inventory. Prior to sharing human research participant data, authors should consult with an ethics committee to ensure data are shared in accordance with participant consent and all applicable local laws.

Response: We haven’t found any data that would compromise participant’s privacy (names, initials, specific dates, contact details, location, etc), therefore have not amended the S1 Complete Dataframe.xlsx file.

5. Please include captions for your Supporting Information files at the end of your manuscript, and update any in-text citations to match accordingly.

Response: We have included a title and caption of the supplementary table at the end of the manuscript.

Response: We have only included references deemed appropriate and relevant.

7. On the previous request concerning identifying data, please remove the fields in the supporting data containing exact ages and retain only the age bins.

Response: Data containing exact ages have been removed from supporting data.

8. Provide further detail in the main text on the consent process to demonstrate that informed consent was appropriately obtained from participants.

Response: We have added an ethics declaration statement in the Methodology section of the manuscript, detailing GDPR compliance and informed consent obtaining.

9. Consistent with best practice and the WMA’s Declaration of Helsinki, I would reiterate the request for the study to be reviewed by an independent ethics panel, or provide clarification if such independent oversight has already taken place, in order to confirm that relevant ethical and legal requirements were followed (e.g. GDPR).

Response:

This study is exempt from the full scope of Switzerland's Human Research Act (HRA, Art. 2), as it does not investigate human diseases, bodily functions, or public health outcomes, nor does it involve any form of physical or psychological intervention. The research consists exclusively of purely observational, anonymous surveys, which fall outside the HRA's regulatory remit. Data collection was conducted in full compliance with the Swiss Federal Act on Data Protection (FADP). We have added an ethics declaration to the manuscript and uploaded a merged document containing consent forms, terms & conditions and privacy policy from Positly alongside this revision.

Comments from Reviewer 1:

1. My one main concern is regarding the conclusion here: "Survey results show that most participants were against making cryopreservation illegal (Fig 2). A total of 83.5% of respondents either opposed or remained neutral toward a ban, with 38.9% explicitly disagreeing or strongly disagreeing and 44.6% expressing neutrality. Only 16.4% agreed or strongly agreed with making it illegal."...

Response: Thank you for pointing this out. This has now been corrected.

2. Also, some of the citations could be formatted better. For example, some of them seem to have authors, but the authors are not listed in the citation. It might also be useful to have a few citations in the beginning to the general topic of cryopreservation/biostasis/cryonics, so that readers know specifically what you are referring to.

Response: We have paid special attention to formatting the citations and added more of them to support our claims and reader’s understanding.

3. One minor point regarding: "This suggests that some individuals may view cryopreservation as a way of coping with death anxiety". I would specifically describe this as a "problem-solving way of coping", rather than just a way of coping. Because theoretically, it's not just about feeling better, which is sometimes the connotation of the word "coping", but rather attempting to solve the problem of death.

Response: Reviewer’s minor point was taken into consideration, and we agree with the proposed idea that cryopreservation is a problem-solving venture. Our data suggests a significant correlation between fear of death and openness to cryopreservation, suggesting a view of cryopreservation way f coping with death anxiety, albeit a solution-oriented one.

Comments from Reviewer 2:

1. Line 1: The title “Cryopreservation sentiment in Switzerland” is too broad. “Cryopreservation” covers embryos, gametes, organs, etc. The survey is specifically about human cryopreservation/cryonics. “Stated attitudes” would be more precise than “Sentiment”. My suggestion for an alternative title is “Swiss public attitudes to cryonics”.

Response: We have taken reviewer’s suggestions into consideration and adapted the title to: Swiss public attitudes to human cryopreservation.

2. Line 8 and 34: The authors describe the 2022 EBF cryonics facility as the first in Europe. The authors should modify this claim, due to the KrioRus facility established in 2005 in geographical Europe. For instance, the authors might replace “Europe” with “Schengen Area”.

Response: Thank you for highlighting this. We have changed both lines from Europe to Schengen Area.

3. Line 26: The authors should discuss and include references to at least some landmark publications on cryonics in bioethical journals, including Shaw Bioethics 2009, Moen JME 2015, Thau Bioethics 2020, as well as more recent article like Sauchelli Sci Eng Ethics 2024, Shao Int J Phil Stud 2025, German and Tretter Neuroethics 2025.

Response: We agree with this point and have expanded the Reference list with more publications in bioethical journals.

4. Figure 1: The authors might consider representing this data as a pie chart or a diverging stacked horizontal bar chart.

Response: we find that the simplicity of representations catches the readers eye and have not modified the figure.

5. Line 128: The authors may also relate their findings regarding family wishes to Hillenbrink R, Wareham CS. Mourning the frozen: considering the relational implications of cryonics. J Med Ethics. 2024

Response: we briefly discussed the mentioned publication in the “Openness to Cryopreservation as Philosophical Stance” section of the Discussion (Page 16, lines 315 – 317).

6. Line 254 and Line 277: If successful, cryonics could have pronounced benefits. It is therefore of general interest to elucidate the underlying psychological and sociological causes for its appeal to the general population in different framings. The manuscript provides some engagement with this topic by pointing to Bandura's reciprocal determinism, but the authors should extend their discussion of potential causal effects.

Response: We have expanded our Openness–Awareness Dynamic discussion section (Page 13, lines 301-304) and Openness to Cryopreservation as Philosophical Stance section (Page 15, lines 334-335), to include potential causal effects.

7. Please clarify competing interests, including affiliations and equity stakes related to Tomorrow Biostasis, the European Biostasis Foundation, and other pertinent entities discussed in the manuscript.

Response: We have added a section on competing interests.

Sincerely,

Emil Kendziorra

On behalf of all authors.

---

## [Editor Report · Decision Letter 2]

25 Mar 2026

Swiss public attitudes to human cryopreservation

PONE-D-25-45890R2

Dear Dr. Kendziorra,

We’re pleased to inform you that your manuscript has been judged scientifically suitable for publication and will be formally accepted for publication once it meets all outstanding technical requirements.

Kind regards,

Barry L. Bentley, Ph.D.

Academic Editor

PLOS One

Additional Editor Comments (optional):

Minor corrects:

(1) Please use the definite article with "Schengen Area", i.e. "the Schengen Area".

(2) Please remove contractions (i.e. change "it's" to "it is").
---

## [Editor Report · Acceptance letter]

PONE-D-25-45890R2

PLOS One

Dear Dr. Kendziorra,

I'm pleased to inform you that your manuscript has been deemed suitable for publication in PLOS One. Congratulations! Your manuscript is now being handed over to our production team.

Kind regards,

on behalf of

Dr Barry L. Bentley

Academic Editor

PLOS One